# Computational Biology and Machine Learning Approaches Identify Rubber Tree (*Hevea brasiliensis* Muell. Arg.) Genome Encoded MicroRNAs Targeting Rubber Tree Virus 1

Muhammad Aleem Ashraf [1,2,*,†] , Hafiza Kashaf Tariq [2,†], Xiao-Wen Hu [3], Jallat Khan [4] and Zhi Zou [1,*]

1   Hainan Key Laboratory for Biosafety Monitoring and Molecular Breeding in Off-Season Reproduction Regions, Institute of Tropical Biosciences and Biotechnology/Sanya Research Institute, Chinese Academy of Tropical Agricultural Sciences, Haikou 571101, China
2   Institute of Biological Sciences, Faculty of Natural and Applied Sciences, Khwaja Fareed University of Engineering and Information Technology, Rahim Yar Khan 64200, Pakistan
3   South Subtropical Crop Research Institute, Chinese Academy of Tropical Agricultural Sciences, Zhanjiang 524000, China
4   Institute of Chemistry, Faculty of Natural and Applied Sciences, Khwaja Fareed University of Engineering and Information Technology, Rahim Yar Khan 64200, Pakistan
*   Correspondence: ashraf.muhammad.aleem@gmail.com (M.A.A.); zouzhi@itbb.org.cn (Z.Z.)
†   These authors contributed equally to this work.

**Abstract:** Tapping panel dryness (TPD), a complex physiological syndrome associated with the rubber tree (*Hevea brasiliensis* Muell. Arg.), causes cessation of latex drainage upon tapping and thus threatens rubber production. Rubber tree virus 1 (RTV1) is a novel positive-sense single-stranded RNA virus from the *Betaflexiviridae* (genus *Capillovirus*), which has been established to cause TPD. MicroRNAs (miRNAs) play an important role in the interplay between viruses and host cells. In this study, we identified the rubber tree genome-encoded miRNAs and their therapeutic targets against RTV1. We applied computational algorithms to predict target binding sites of rubber tree miRNAs potentially targeting RTV1 RNA genome. Mature rubber-tree miRNAs are retrieved from the miRBase database and are used for hybridization of the RTV1 genome. A total of eleven common rubber-tree miRNAs were identified based on consensus genomic positions. The consensus of four algorithms predicted the hybridization sites of the hbr-miR396a and hbr-miR398 at common genomic loci (6676 and 1840), respectively. A miRNA-regulatory network of rubber tree was constructed with the RTV1—ORFs using Circos, is illustrated to analyze therapeutic targets. Overall, this study provides the first computational evidence of the reliable miRNA–mRNA interaction between specific rubber tree miRNAs and RTV1 genomic RNA transcript. Therefore, the predicted data offer valuable evidence for the development of RTV1-resistant rubber tree in the future. Our work suggests that similar computational host miRNA prediction strategies are warranted for identification of the miRNA targets in the other viral genomes.

**Keywords:** rubber tree capillovirus 1; microRNAs; plant–virus interaction; RNAi; computational algorithms; gene silencing; minimum free energy

## 1. Introduction

Rubber tree (*Hevea brasiliensis* Muell. Arg., Euphorbiaceae, Malpighiales) is a valuable source of natural rubber (NR) which is an indispensable commercial source material for the manufacturing of more than 5000 products worldwide [1]. The rubber tree genome contains a diploid set of 36 chromosomes, and the first draft genome was released in 2013 [2]. The production of natural rubber is highly hampered by different pathogens that infect the rubber tree [3–7]. Recently, a novel capillovirus was identified infecting the rubber tree in China. The RNA genome of the novel capillovirus was isolated, sequenced and was tentatively assigned the nomenclature of rubber tree virus 1 (RTV1). RTV1 has emerged as

a deleterious pathogen of the genus *Capillovirus* in the family *Betaflexiviridae*. RTV1 consists of a monopartite, linear, non-enveloped, +ssRNA molecule of 6811 nucleotides [6].

Considering the RNA nature of RTV1, RNA-based molecular approaches, RNA interference (RNAi) and host-derived microRNA (miRNA) silencing are emerging as the potential tools, used to cleave the viral mRNA in the infected host cells. RNA silencing is a conserved, sequence-specific gene silencing mechanism controlled by the siRNAs [8]. It is an important line of defense against invading viruses in the host cell [9]. The Dicer and Argonaute genes are the key components of the RNAi machinery. The cleavage of the precursor dsRNA results in short 21–24 nucleotides siRNA that inhibit protein translation during infection [10]. The artificial microRNA (amiRNA)-mediated RNAi has high silencing specificity and develops a single 21-nucleotide amiRNA to target the corresponding sequence [11]. The plant miRNAs are small endogenous, noncoding, regulatory bigwigs of 20–24 nucleotides in length which are encoded by *MIR* genes [12]. They can regulate complex biological processes including gene expression and regulations in plants [13]. The rubber tree has evolved diverse molecular mechanisms and has inherited mature miRNAs that provide immunity against biotic and abiotic stresses [14,15].

The amiRNA-mediated silencing technology has emerged as a novel approach to control plant viruses. In a previous study, the amiRNA-based construct was designed and transformed in Arabidopsis to create resistance against *Tymovirus* and *Potyvirus* [11]. In the rubber tree genome, 30 mature miRNAs have been found [16]; a subset of these mature miRNAs in the rubber tree should have targets in the RTV1 genome. This current computational biology approach was based on a comprehensive bioinformatics analysis of the RTV1 genome using the rubber tree miRNAs. The current study aims to implement computational algorithms for the prediction of the rubber tree genome-encoded miRNAs targeting RTV1. The predicted rubber tree miRNAs can be utilized for the development of the amiRNA-based constructs to transform in the rubber tree to control the RTV1 in the future.

## 2. Materials and Methods

### 2.1. Retrieval of Rubber Tree MicroRNAs

A total of 30 mature rubber tree (commonly called *Hevea brasiliensis*) microRNAs (hbr-miR156-hbr-miR9387) (Accession IDs: MIMAT0025282-MIMAT0035236) (Table S1 (Supplementary Materials)) and stem-loop hairpin precursor miRNAs (hbr-MIR156-hbr-MIR9387) (Accession IDs: MI0022052-MI0028936) (Table S2) were downloaded from the miRBase version 22 (http://mirbase.org/) (accessed on 26 October 2021) biological database [16].

### 2.2. RTV1 Genome Retrieval and Annotation

The full-length genomic transcript of the RTV1 (Accession ID: MN047299) was retrieved from the National Centre for Biotechnology Information (NCBI) GenBank database. Annotation and production of the graphical output of the RTV1 ORFs was created by the pDRAW32 DNA plasmid map analysis (AcaClone 1.1.147) software.

### 2.3. miRanda

The miRanda is a seed-based standard computational scanning algorithm. It was implemented for the first time in 2003 [17]. It has been updated into a web-based tool (http://www.microrna.org/) (accessed on 9 June 2022) for the prediction and analysis of miRNA [18]. It was run under well-defined standard settings. The following parameters were set for analysis: gap open penalty: (−8.00); gap extend: (−2.00); score threshold: (50.00); minimum free energy (MFE) threshold: (−20.00 Kcal/mol) and scaling parameter: (2.00).

### 2.4. RNA22

RNA22 is a non-seed-based user-friendly, pattern-recognition algorithm that was assessed at http://cm.jefferson.edu/rna22v1.0/ (accessed on 9 November 2021) [19]. It

has been developed for the identification of target binding sites of miRNAs and their interrelated heteroduplexes. Characteristic parameters were set as the following: output format (heteroduplexes); sensitivity vs. specificity setting: (sensitivity (63%), specificity (61%)); seed region: (seed size of 7, allow maximum of I UN-paired bases in seed); minimum number of paired-up bases in heteroduplex: (12), maximum folding energy for heteroduplex: ($-16.00$ Kcal/mol) and Maximum number of G: U wobbles allowed in seed region: (no limit).

### 2.5. RNAhybrid

RNAhybrid is a seed-based intermolecular hybridization algorithm which is used to predict miRNA targets in a very easy and flexible manner. It is an online available tool for the rapid prediction of miRNA targets based on MFE hybridization of mRNA and miRNAs [20]. Salient parameters were set as the following: MFE threshold ($-20$ Kcal/mol); hit per target (1); no G: U in seed; helix constraint from and helix constraint to.

### 2.6. Tapirhybrid

The Tapirhybrid is a webserver used to predict plant miRNA targets as well as target mimics using a fast and a precise algorithm [21]. Tapirhybrid was run in FASTA mode using the following settings: a score cutoff score <= 9 and minimum free energy (MFE) ratio: (0.200).

### 2.7. psRNATarget

The psRNATarget program is a newly designed browser-based tool (http://plantgrn.noble.org/psRNATarget/) (accessed at 9 November 2021) for the prediction of plant miRNA targets. Complementary scoring and secondary structure prediction are the key features of the psRNATarget algorithm [22]. The following standard key features were set: no. of top targets (200); expectation score (5); mode of inhibition (cleavage); penalty for G: U pair: (0.5); penalty for other mismatches: (1); seed region: 2–13 (NT); no. of mismatches allowed in seed region: (2); HSP size: (19); penalty for opening gap: (2); and penalty for extending gap: (0.5).

### 2.8. Mapping of miRNA–Target Interaction

The miRNA–target interaction was mapped using the Circos algorithm [23]. An interaction map was developed using rubber tree miRNAs and RTV1 genes.

### 2.9. RNAfold and RNAcofold

RNAfold is one of the latest web-based algorithms used for the prediction of secondary structures from the precursor sequences using MFE as standard. Analysis was performed with user-defined settings (MFE, partition function, avoid isolated base pairs and interactive RNA secondary structure plot) [24]. RNAcofold is a web-based newly developed algorithm used to estimate the free energy ($\Delta$G) of duplex binding. It is used to evaluate the mRNA and miRNA duplex interaction. Consensus rubber tree miRNAs and corresponding RTV1 target region sequences were processed under default parameters [25].

### 2.10. Graphical Representation

R (version 3.1.1) studio was used to process all the biological data into graphical representations, and readxl and ggplot2 were implemented in generating the graphical output [26].

## 3. Results

### 3.1. Rubber Tree miRNA's Loci on RTV1 Genome

A biological computational framework was designed using the plant miRNA prediction algorithms and host-derived miRNAs from the miRBase (Figure 1). We investigated the possibility of the rubber tree (host) miRNAs with a potential to target the RTV1 genome.

We generated the RTV1 genome from the NCBI database and the computational annotation of different ORFs was performed (Figure 2). We employed five different miRNA prediction algorithms (miRanda, RNA22, RNAhybrid, Tapirhybrid and psRNATarget) to predict the rubber tree miRNA binding strength in the genome of the RTV1 ssRNA molecule. The miRanda algorithm has predicted 12 rubber tree miRNAs targeting 12 loci. RNA22: 6 rubber tree miRNAs targeting 8 loci. RNAhybrid: 29 rubber tree miRNAs targeting 29 loci. Tapirhybrid has predicted that 5 rubber tree miRNAs targeted 5 loci. psRNATarget: 14 rubber tree miRNAs targeting 23 loci (Figure 3) (Table S3).

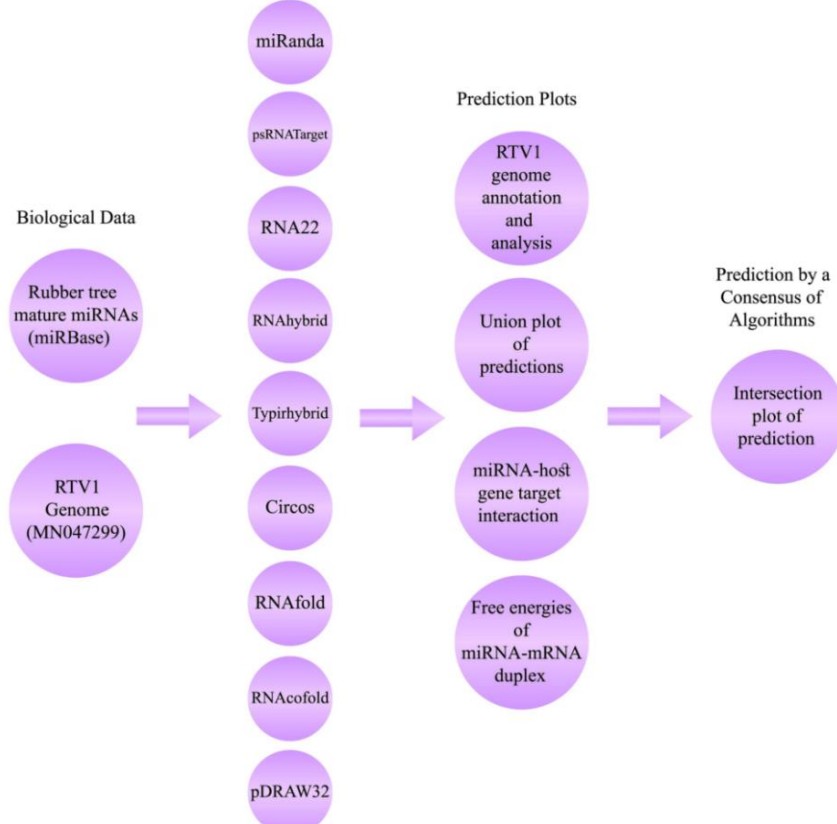

**Figure 1.** A computational biology scheme and methodology of rubber tree miRNA prediction from the RTV1 genome.

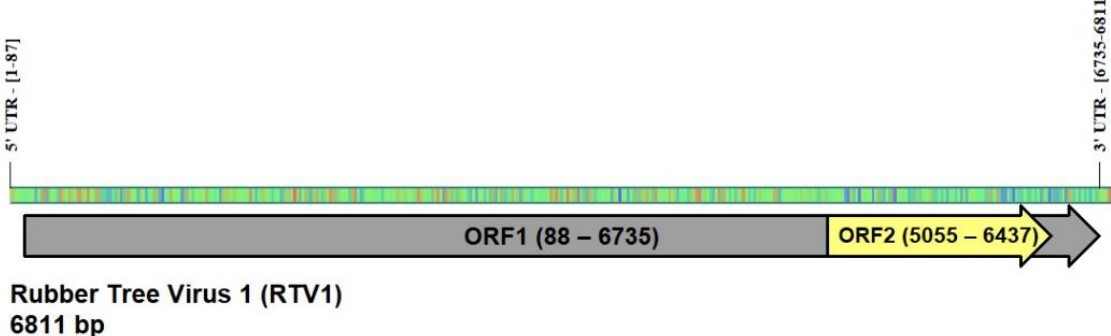

**Figure 2.** Schematic representation of the RTV1 genome. Coordinates are designed based on accession number of the RTV1 genome.

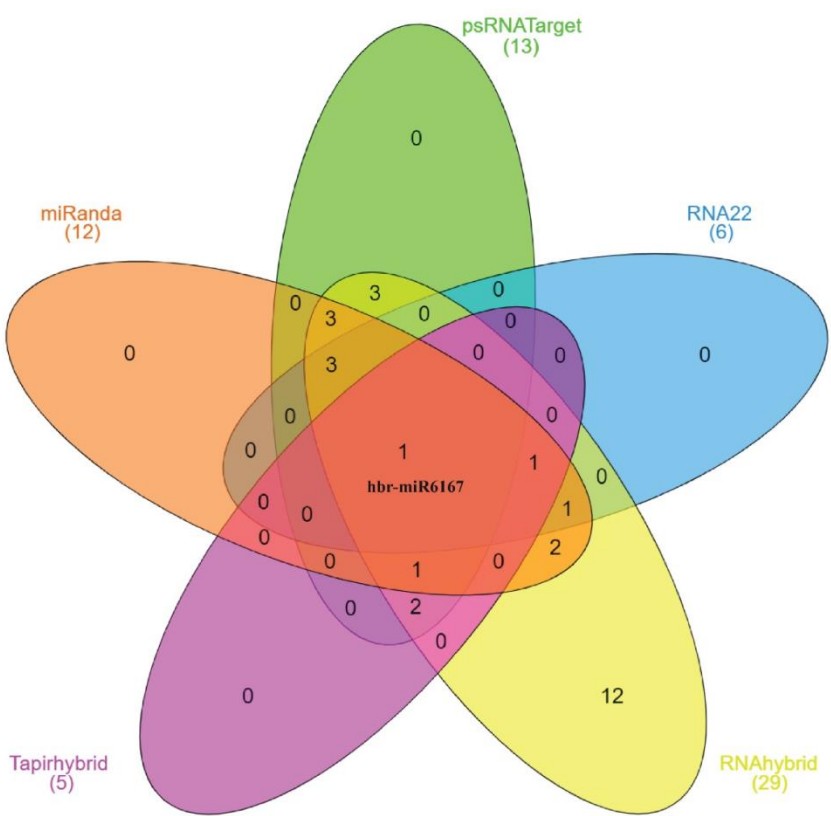

**Figure 3.** Venn diagram plot of RTV1 representing common rubber tree genome-encoded miRNAs calculated by all the algorithms used in this study. Furthermore, a single rubber tree miRNA (hbr-miR6167) is predicted by all the algorithms used in this study.

### 3.2. ORF1 Encoding Polyprotein

ORF1 (88–6735 bp) encodes a polyprotein (2215 AA) of the RTV1 genome. ORF1 was targeted by seven rubber tree miRNAs as indicated by the miRanda algorithm: hbr-miR159a (locus 2178); hbr-miR319 (locus 2992); hbr-miR396 (a, b) (locus 6678); hbr-miR398 (locus 1839); hbr-miR408b (locus 6498); and hbr-miR6171 (locus 1633) (Figure 4a). Six potential miRNAs were identified for silencing the ORF1of the RTV1 genome by RNA22: hbr-miR319 (loci 569, 3204); hbr-miR396a locus (6675); hbr-miR398 (locus 1838); hbr-miR482a (locus 3951); hbr-miR6167 (locus 1058); and hbr-miR6168 (loci 145, 647) (Figure 4b). Twenty two miRNAs were detected by the RNAhybrid : hbr-miR156 (locus 6492); hbr-miR159a (locus 2177); hbr-miR166a (locus 1060), hbr-miR396a (locus 6675), hbr-miR396b (locus 823), hbr-miR398 (locus 1840); hbr-miR408a (locus 573); hbr-miR408b (locus 6497); hbr-miR482b (locus 6500); hbr-miR2118 (locus 6592); hbr-miR6166 (locus 1515); hbr-miR6168 (locus 1641); hbr-miR6169 (locus 3882); hbr-miR6170 at (locus 6660); hbr-miR6171 (locus 2043); hbr-miR6172 (locus 4500); hbr-miR6173 (locus 3749); hbr-miR6174 (locus 4303); hbr-miR6482 (locus 2163); hbr-miR6483(locus 1267); hbr-miR6485 (locus 3042); and hbr-miR9386 (locus 370) (Figure 4c). Tapirhybrid predicted only four miRNAs: hbr-miR396a (locus 6674); hbr-miR6167 (locus 2716); hbr-miR6171 (locus 1633); and hbr-miR6483 at (locus 106) (Figure 4d).

The psRNATarget identified twelve miRNAs: hbr-miR159a (loci 2177, 3505, 2549, 4786); hbr-miR319 (locus 2548); hbr-miR396b (locus 6677); hbr-miR398 (locus 1838); hbr-miR482a (locus 627); hbr-miR6167 (loci 1459, 980); hbr-miR6169 (locus 1693); hbr-miR6171 (locus 1633, 4540); hbr-miR6482 (locus 399); hbr-miR6483 (locus 106, 3183); hbr-miR6484 (locus 4295); and hbr-miR9386 locus (4422) (Figure 4e,f; Tables 1, S3 and S4; File S1).

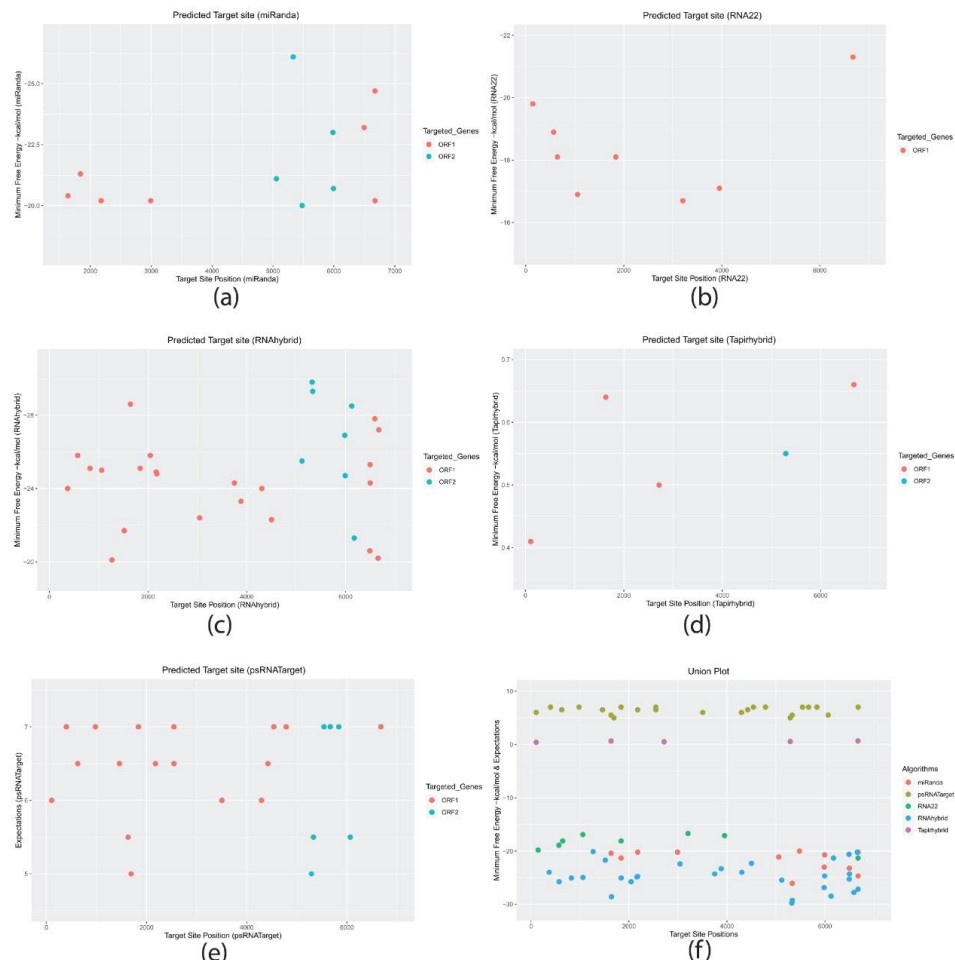

**Figure 4.** miRNA target prediction results of rubber tree virus 1. Five miRNA target prediction algorithmic tools, including (**a**) miRanda, (**b**) RNA22, (**c**) RNAhybrid, (**d**) Tapirhybrid and (**e**) psR-NATarget, were used in this study. (**f**) miRNA prediction results represented as a union of multiple algorithms.

**Table 1.** Rubber tree miRNAs and their target binding sites in the RTV1 genome predicted by different algorithms in this study.

| Rubber Tree miRNAs | Locus miRanda | Locus RNA22 | Locus RNAhybrid | Locus TAPIR | Position psRNATarget | MFE * miRanda | MFE ** RNA22 | MFE * RNAhybrid | MFE-Ratio TAPIR | Expectation psRNATarget |
|---|---|---|---|---|---|---|---|---|---|---|
| hbr-miR156 | | | 6492 | | | | | −20.6 | | |
| hbr-miR159a | 2178 | | 2177 | | 2177 | −20.2 | −15.2 | −24.8 | | 6.5 |
| hbr-miR159a(1) | | | | | 3505 | | | | | 6 |
| hbr-miR159a(2) | | | | | 2549 | | | | | 6.5 |
| hbr-miR159a(3) | | | | | 4786 | | | | | 7 |
| hbr-miR166a | | | 1060 | | | | | −25 | | |
| hbr-miR166b | | | 6125 | | | | | −28.5 | | |
| hbr-miR319 | 2992 | 569 | 5323 | | 2548 | −20.2 | −18.9 | −29.8 | | 7 |
| hbr-miR319(1) | | 3204 | | | | | −16.7 | | | |
| hbr-miR396a | 6676 | 6675 | 6675 | 6674 | | −24.7 | −21.3 | −27.2 | 0.66 | |
| hbr-miR396b | 6678 | | 823 | | 6677 | −20.2 | | −25.1 | | 7 |
| hbr-miR396b(1) | | | | | 5836 | | | | | 7 |
| hbr-miR398 | 1839 | 1838 | 1840 | | 1838 | −21.3 | −18.1 | −25.1 | | 7 |
| hbr-miR408a | | | 573 | | | | −14.2 | −25.8 | | |
| hbr-miR408b | 6498 | | 6497 | | | −23.2 | −14.9 | −25.3 | | |
| hbr-miR476 | | | 6176 | | | | | −21.3 | | |
| hbr-miR482a | 5332 | 3951 | 5334 | | 5331 | −26.1 | −17.1 | −29.3 | | 5.5 |
| hbr-miR482a(1) | | | | | 627 | | | | | 6.5 |
| hbr-miR482b | | | 6500 | | | | | −24.3 | | |
| hbr-miR2118 | 5480 | | 6592 | | | −20 | | −27.8 | | |
| hbr-miR6166 | | | 1515 | | | | | −21.7 | | |
| hbr-miR6167 | 5985 | 1058 | 5984 | 2716 | 1459 | −23 | −16.9 | −26.9 | 0.5 | 6.5 |

**Table 1.** *Cont.*

| Rubber Tree miRNAs | Locus miRanda | Locus RNA22 | Locus RNAhybrid | Locus TAPIR | Position psRNATarget | MFE * miRanda | MFE ** RNA22 | MFE * RNAhybrid | MFE-Ratio TAPIR | Expectation psRNATarget |
|---|---|---|---|---|---|---|---|---|---|---|
| hbr-miR6167(1) | | | | | 980 | | | | | 7 |
| hbr-miR6168 | 5056 | 145 | 1641 | | | −21.1 | −19.8 | −28.6 | | |
| hbr-miR6168(1) | | 647 | | | | | −18.1 | | | |
| hbr-miR6169 | | | 3882 | 5291 | 5291 | | −14.5 | −23.3 | 0.55 | 5 |
| hbr-miR6169(1) | | | | | 1693 | | | | | 5 |
| hbr-miR6170 | | | 6660 | | | | | −20.2 | | |
| hbr-miR6171 | 1633 | | 2043 | 1633 | 1633 | −20.4 | −15.8 | −25.8 | 0.64 | 5.5 |
| hbr-miR6171(1) | | | | | 6066 | | −15.8 | | | 5.5 |
| hbr-miR6171(2) | | | | | 4540 | | −15.8 | | | 7 |
| hbr-miR6172 | | | 4500 | | | | −15.3 | −22.3 | | |
| hbr-miR6173 | | | 3749 | | | | | −24.3 | | |
| hbr-miR6174 | | | 4303 | | 5542 | | | −24 | | 7 |
| hbr-miR6175 | | | 5118 | | | | −14.9 | −25.5 | | |
| hbr-miR6482 | | | 2163 | | 399 | | | −24.9 | | 7 |
| hbr-miR6483 | | | 1267 | 106 | 106 | | | −20.1 | 0.41 | 6 |
| hbr-miR6483(1) | | | | | 5666 | | | | | 7 |
| hbr-miR6484 | 5994 | | 5993 | | 4295 | −20.7 | | −24.7 | | 6 |
| hbr-miR6485 | | | 3042 | | | | | −22.4 | | |
| hbr-miR9386 | | | 370 | | 4422 | | | −24 | | 6.5 |
| hbr-miR9387 | | | | | | | | | | |

MFE *: minimum free energy; MFE **: minimum folding energy.

### 3.3. ORF2 Encoding Movement Protein

Movement protein (MP, 460AA) is encoded by the ORF2 (5055–6437 bp). The miRanda predicted five miRNAs: hbr-miR482a (locus 5332); hbr-miR2118 (locus 5480); hbr-miR6167 (locus 5985); hbr-miR6168 (locus 5056); and hbr-miR6484 (locus 5994) (Figure 4a). No rubber tree miRNAs were identified to target the ORF2 gene with the RNA22 algorithm (Figure 4b). RNAhybrid predicted the following miRNAs: hbr-miR166b (locus 6125); hbr-miR319 (locus 5323); hbr-miR476 (locus 6176); hbr-miR482a (locus 5334); hbr-miR6167 (locus 5984); hbr-miR6175 (locus 5118); and hbr-miR6484 (locus 5993) (Figure 4c). Tapirhybrid predicted the binding of hbr-miR6169 at locus 5291 (Figure 4d). The psRNATarget predicted six miR-NAs: hbr-miR396b (locus 5836); hbr-miR482a (locus 5331); hbr-miR6169 (locus 5291); hbr-miR6171 (locus 6066); hbr-miR6174 (locus 5542); and hbr-miR6483 (locus 5666) (Figure 4e,f; Tables 1, S3 and S4; File S1).

### 3.4. Visualization of miRNA–Target Interaction Network

Circos plotting is widely used to understand host–virus interaction using the biological data in a precise manner. The mapped consensual rubber tree miRNAs are depicted in the RTV1 genomic ORFs (Figure 5).

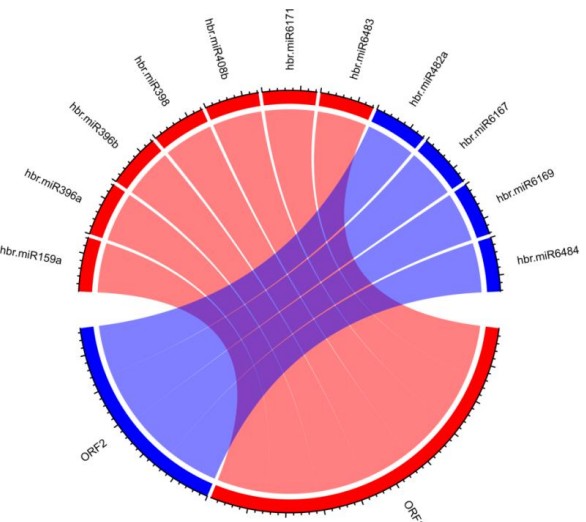

**Figure 5.** A schematic interaction Circos map representing the rubber tree miRNAs and RTV1 ORFs. The colored lines represent the RTV1 genomic components (ORF1-2).

### 3.5. Predicting Common Rubber Tree miRNAs

Based on predicted targeting miRNAs from rubber tree to silence the RTV1 genome, the best single miRNA (hbr-miR6167) was detected by union of consensus between multiple algorithms (miRanda, RNA22, RNAhybrid, Tapirhybrid and psRNATarget) (Figure 3). Furthermore, three rubber tree miRNAs (hbr-miR482a, hbr-miR398 and hbr-miR319), were predicted by union of consensus between multiple algorithms (miRanda, RNA22, RNAhybrid and psRNATarget). Three common rubber tree miRNAs (hbr-miR396b, hbr-miR159a and hbr-miR6484), were detected by at least three algorithms used in this study (Figure 3; Table 1).

### 3.6. Prediction of Consensual Rubber Tree miRNAs

Of the 30 targeting mature rubber tree miRNAs, only 2 rubber tree miRNAs (hbr-miR396a locus (6674) and hbr-miR398 locus (1838)) were predicted by union of consensus between multiple algorithms (Figure 6). Furthermore, three consensus miRNAs (hbr-miR159a, hbr-482a and hbr-miR6171) at unique positions (2177, 5331 and 1633, respectively) were predicted to have potential target-binding sites at the common locus; this was confirmed by three algorithms. Interestingly, seven consensual rubber tree miRNAs (hbr-miR396b, hbr-miR408b, hbr-miR6167, hbr-miR6169, hbr-miR6483 and hbr-miR6484) at unique positions (6677, 6497, 5984, 5291, 106 and 5993, respectively), showed consensus unique hybridization binding sites at the common locus; these were confirmed by two algorithms (Figure 6; Table S3).

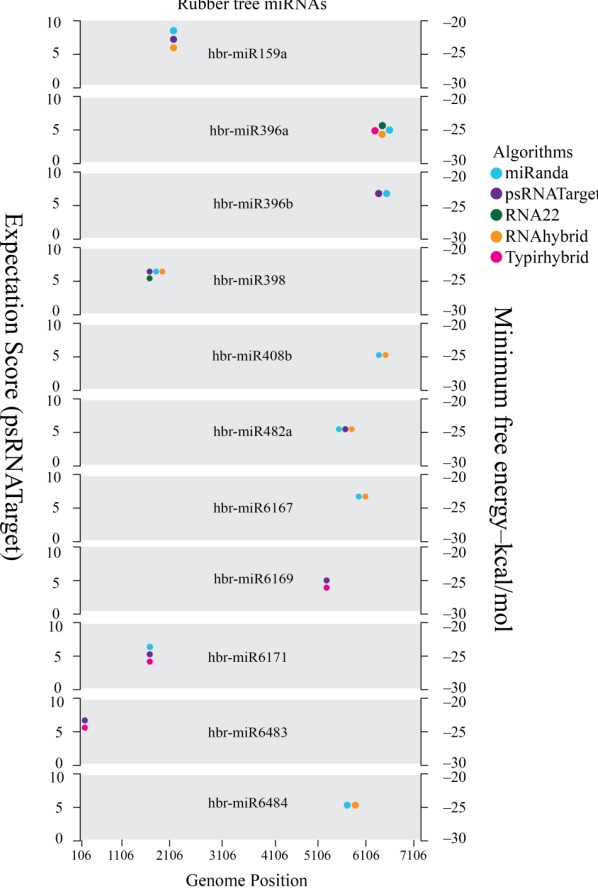

**Figure 6.** Intersection plot representing consensual rubber tree miRNAs predicted by different algorithms at common locus positions. Minimum free energy (MFE) for potential consensual miRNAs is considered −20 Kcal/mol by at least two of the algorithms used in this study. MFE (miRanda, RNA22 and RNAhybrid) and expectation cut-off score (psRNATarget) are indicated. Color codes are given within the figure.

### 3.7. Prediction of Consensual Secondary Structures

The validation of the predicted consensual rubber tree miRNAs was monitored by production of their secondary structures from the precursor sequences. The precursors of mature rubber tree miRNAs were manually curated. The MFE is the key factor to determine the stable secondary structures of precursors. In this study, the significant parameters of eleven consensual secondary structures of precursor miRNAs were also identified, such as length miRNA, length precursor, MFE, GC content, AMFE and MFEI. In our studies, the length precursor ranges from 86–221 nucleotides, along with MFE (−16.50 to −101.30 Kcal/mol), GC content of 27–57%, AMFE of −13.86 to −46.50 and MFEI ranges from −0.49 to −1.26 (Figure 7; Tables 2 and S5).

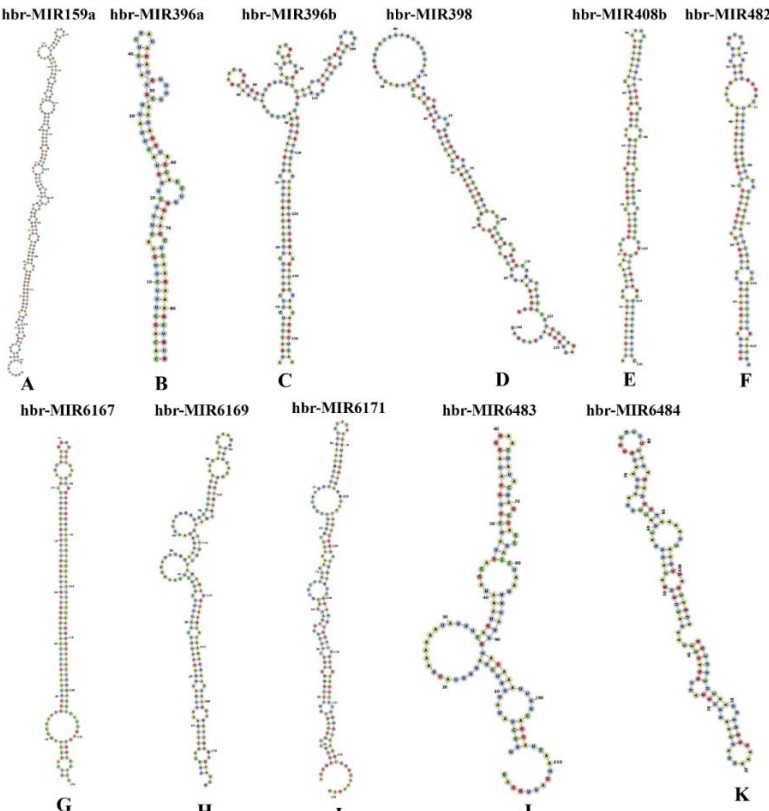

**Figure 7.** Stable secondary structures of eleven consensual precursors of the mature rubber tree miRNAs identified. Names with MFE are indicated: (**A**) hbr-MIR159a (−89.80 Kcal/mol), (**B**) hbr-MIR396a (−35.40 Kcal/mol), (**C**) hbr-MIR396b (−64.20 Kcal/mol), (**D**) hbr-MIR398 (−42.80 Kcal/mol), (**E**) hbr-MIR408b (55.80 Kcal/mol), (**F**) hbr-MIR482a (−52.00 Kcal/mol), (**G**) hbr-MIR6167 (−101.30 Kcal/mol), (**H**) hbr-MIR6169 (−33.90 Kcal/mol), (**I**) hbr-MIR6171 (−43.50 Kcal/mol), (**J**) hbr-MIR6483 (−16.50 Kcal/mol) and (**K**) hbr-MIR6484 (−23.10 Kcal/mol).

**Table 2.** The parameters of consensus secondary structures of precursor miRNAs.

| miRNA ID | Length of miRNA | Length of Precursor | MFE * (Kcal/mol) | AMFE ** | MFEI *** | (G + C)% | ΔG **** (Kcal/mol) |
|---|---|---|---|---|---|---|---|
| hbr-miR396a | 21 | 86 | −35.40 | −41.16 | −1.14 | 36 | −22.50 |
| hbr-miR398 | 21 | 140 | −42.80 | −30.57 | −0.61 | 50 | −18.80 |

* MFE means Minimum Free Energy, ** AMFE means Adjusted Minimum Folding Free Energy, *** MFEI means Minimum Folding Free Energy Index, **** ΔG means minimum free energy of duplex formation. AMFE = MFE/Length of a pre-miRNA × 100, MFEI = (MFE/Length of a pre-miRNA × 100)/(G + C)%.

### 3.8. Evaluation of Free Energy (ΔG) of mRNA-miRNA Interaction

The predicted consensual rubber tree miRNAs were evaluated by calculating the free energies (ΔG) of the miRNA/target duplexes. Eleven consensual rubber tree miR-

NAs were used to estimate the ($\Delta G$). These were hbr-miR159a ($\Delta G$: $-18.90$ Kcal/mol), hbr-miR396a ($\Delta G$: $-22.50$ Kcal/mol), hbr-miR396b ($\Delta G$: $-19.40$ Kcal/mol), hbr-miR398 ($\Delta G$: $-18.80$ Kcal/mol), hbr-miR408b ($\Delta G$: $-19.9.50$ Kcal/mol), hbr-miR482a ($\Delta G$: $-24.20$ Kcal/mol), hbr-miR6167 ($\Delta G$: $-21.10$ Kcal/mol), hbr-miR6169 ($\Delta G$: $-16.70$ Kcal/mol), hbr-miR6171 ($\Delta G$: $-19.20$ Kcal/mol), hbr-miR6483 ($\Delta G$: $-11.80$ Kcal/mol) and hbr-miR6484 ($\Delta G$: $-17.50$ Kcal/mol) (Table S5).

## 4. Discussion

We studied the efficiency of different computational algorithms used here, to evaluate the predicted target binding sites of rubber tree miRNAs for screening of false positive findings. In the RTV1 viral genome, predictions based on computational algorithms provide quick ways to identify potential host-derived miRNAs. At the individual, intersection and union levels, we developed the most effective approach for analyzing rubber tree miRNA prediction findings. Using several prediction algorithms, default algorithm characteristic parameters were efficient for filtering out false-positive targets [27,28]. Default algorithms indicate optimum identification for a miRNA to its appropriate viral genome target site [28]. In the current study, several potential mature rubber tree miRNAs (hbr-miR159a, hbr-miR396 (a, b), hbr-miR398, hbr-miR408b, hbr-miR482a, hbr-miR6167, hbr-miR6169, hbr-miR6171, hbr-miR6483 and hbr-miR6484) were selected to generate a RTV1-resistant variety. The study indicated that hbr-miR396a and hbr-miR398 were selectively employed by RTV1. Computational algorithms (miRanda, RNA22, RNAhybrid and Tapirhybrid) predicted the consensus hybridization site of hbr-miR396a at locus (6676), while consensus of four algorithms predicted the target binding site of hbr-miR398 at a common locus, 1840. Intriguingly, the MFEs of the consensus target pairs were calculated as very high (Figures 4 and 6; Table 1).

The RNA duplex is considered to be more stable due to the stronger binding of miRNA to mRNA [29–32]. The miRanda algorithm was used to test a variety of factors, ranging from target site preservation to miRNA target prediction across the entire genome. Conservation level is the key feature of the miRanda algorithm [33]. Meanwhile, the psRNATarget algorithm was utilized to predict miRNA target binding sites, using unique plant-based features [22]. The RNA22 algorithm was distinct from the other algorithms due to its unique feature which is pattern-based recognition of miRNA–mRNA interactions [19]. RNAhybrid was implemented for the calculation of minimum free energy (MFE) and used to evaluate target inhibition of rubber tree miRNAs as per the Broderson conclusion [20,34]. The intersection of the psRNATarget and Tapirhybrid algorithms' output delivers an extremely precise prediction of rubber tree miRNAs in this study [21,33].

In this study, we designed three algorithmic approaches at individual, union and intersection levels to evaluate false-positive prediction results. The union approach depends upon the combination of one or more prediction tools to predict the false positive results. This approach increases the sensitivity level of prediction as compared to the specificity of the results; meanwhile, the intersection approach is entirely reliant on the combination of algorithmic tools and enhances the specificity of the prediction results [35]. Our target prediction results revealed that both computational approaches achieved the best outcomes with maximum performance for predicting and estimating the best targets (Figures 3 and 6). Previous experimental studies have also demonstrated the silencing of plant viruses using host genome-encoded miRNAs [11,36–40].

In the current study, predicted consensual miRNAs were concluded after setting the standard setting parameters of the algorithms: MFE; seed pairing; target site accessibility; minimum folding energy (MFE) and pattern recognition. Thus, integrating all major aspects of miRNA target prediction was considered during prediction. Therefore, these predicted miRNAs are considered as the most effective selections for the silencing of the RTV1 genome. Rubber tree miRNA target binding-sites in the RTV1 genome and miRNA–mRNA duplex was consensually identified by all five algorithms.

In this study, the RTV1 genome (ORF1 and ORF2) is vulnerable to targeting by eleven consensus rubber tree miRNAs (Figure 6). Out of eleven predicted consensus rubber tree miRNAs, hbr-miR398 was identified to target ORF1 by four computational algorithms such as miRanda, psRNATarget, RNA22 and RNAhybrid (Figure 6; Table 1 and Table S4). We predicted the evaluation of the free energy of mRNA–miRNA duplex using RNAcofold. The free energy of miRNA–mRNA duplex binding is highly stable in this study.

In the previous studies on host–virus interaction, in silico algorithms were used to study host genome-encoded miRNA targets against potato virus Y (PVY) [35], zucchini yellow mosaic virus (ZYMV) [41], cotton leaf curl Kokhran virus-Burewala (CLCuKoV-Bu) [30], rice yellow mottle virus (RYMV) [42], maize chlorotic mottle virus (MCMV) [43], RTV1 [44], Physostegia chlorotic mottle virus (PhCMoV) and tomato brown rugose fruit virus (ToBRFV) [45]. We have designed similar studies against sugarcane viruses using computational algorithms to understand the host–virus interaction as well as finding the best target in our previous research [31,46,47]. We predicted that hbr-miR396a and hbr-miR398 would be an excellent choice for targeting the RTV1 genome in this work. To establish RTV1 replication experimentally, it is critical to evaluate the function of predicted consensual rubber tree miRNAs for the detection of *Capillovirus* replication. The designed amiRNA construct has high specificity to the target gene to control the detrimental off-target effects. Therefore, the silencing expression can be stably transformed to future generations [48]. Furthermore, the small size of amiRNA permits the insertion of multiple and distinct amiRNAs within a single gene expression cassette, which can then be transformed to develop transgenic plants that are simultaneously resistant to multiple viruses [49]. An amiRNA-based construct was designed to combat RTV1 in rubber tree cultivars that harbor a modified miRNA/miRNA* sequence in a duplex of the precursor (hbr-MIR396a and hbr-MIR398), as shown in Figure 8. Our computational study on RTV1 genome silencing might pave the way for a novel approach to the antiviral effects of host-derived miRNAs against RTV1.

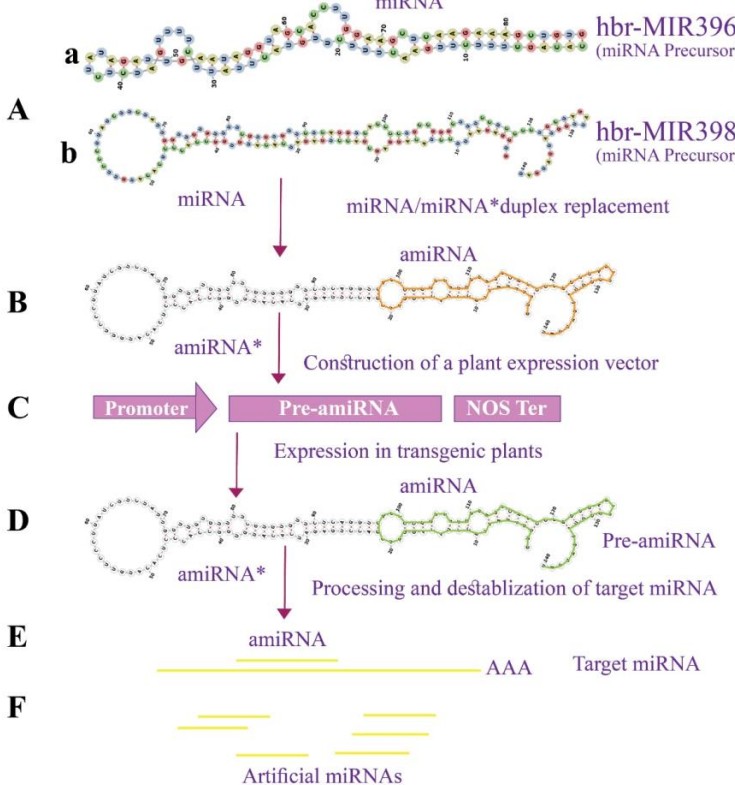

**Figure 8.** Schematic representation of amiRNA-mediated gene-silencing strategy designed to develop transgenic rubber tree cultivars. (**A**) representing the candidate consensus cassava precursors (a) hbr-MIR396a and (b) hbr-MIR398. (**B**) showing designed amiRNA after miRNA/miRNA duplex replacement.

(**C**) showing gene expression cassette containing precursor sequence controlled by a 5′ promoter and NOS terminator is cloned at 3′ end. (**D**) showing pre-amiRNAs processed to develop a mature amiRNA/amiRNA*duplexes. (**E**) representing processing of amiRNAs by RNA induced silencing complex (RISC). (**F**) showing amiRNAs get incorporated to be released for mRNA translation inhibition or degradation of mRNA.

## 5. Conclusions and Recommendations

RTV1 has emerged as a major problem for rubber production in China. Since the application of RNAi-based gene-silencing technology, many plant research laboratories around the world have investigated the silencing of genes using host-encoded miRNAs against plant viruses. In this study, prior to cloning, we have applied tools and computational pipelines to identify candidate miRNA from the rubber tree against RTV1. Among the 11 best consensual candidate miRNAs targeting RTV1, hbr-miR396a and hbr-miR398 were predicted as the most potent miRNAs to target the RTV1 genome. Pathological studies are required to validate the rubber tree miRNA–RTV1 genomic interaction. Our results demonstrate an alternative strategy to existing molecular approaches that could be repurposed to control RTV1 infections. The designed amiRNA constructs are highly suitable for the development of RTV1-resisitant transgenic rubber tree plants after rubber tree transformation in the future.

**Supplementary Materials:** The following supporting information can be downloaded at: https://www.mdpi.com/article/10.3390/app122412908/s1: Table S1: Rubber tree mature microRNAs retrieved from miRBase; Table S2: Rubber tree precursors or pre-miRNAs retrieved from miRBase caption; Table S3: Prediction of effective target binding sites of rubber tree miRNAs in the RTV1 genome using computational algorithms; Table S4: Gene wise prediction by different algorithms; Table S5: The significant parameters of consensus secondary structures of precursor miRNAs; and File S1: Prediction results of different algorithms in the RTV1 genome caption.

**Author Contributions:** M.A.A. and Z.Z. conceived the original idea of the work and designed the study. All the authors performed, analyzed and interpreted the in silico data and wrote the manuscript. All authors have read and agreed to the published version of the manuscript.

**Funding:** M.A.A. was partially financed by Talented Young Scientist Program of China, the National Natural Science Foundation of China (31971688), and the Central Public-interest Scientific Institution Basal Research Fund for Chinese Academy of Tropical Agricultural Sciences (1630052022001).

**Institutional Review Board Statement:** Not applicable.

**Informed Consent Statement:** Not applicable.

**Data Availability Statement:** Not applicable.

**Acknowledgments:** The authors acknowledge Khwaja Fareed University of Engineering and Information Technology for providing financial support to M.A.A. and H.K.T.

**Conflicts of Interest:** The authors declare no conflict of interest.

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
