# Peer review of "Computational Biology and Machine Learning Approaches Identify Rubber Tree (Hevea brasiliensis Muell. Arg.) Genome Encoded MicroRNAs Targeting Rubber Tree Virus 1"

_applsci, doi:10.3390/app122412908_

Round 1

Reviewer 1 Report

The Manuscript [applsci-2010697] entitled (Computational Biology and Machine Learning Approaches to Identify the Rubber Tree (Hevea brasiliensis Muell.Arg.) Genome Encoded MicroRNAs Targeting Rubber Tree Virus 1) provides a first computational evidence of the reliable miRNA–mRNA interaction between specific rubber tree miRNAs and RTV1 genomic RNA transcript and offer valuable evidence for the development of RTV1-resistant rubber tree plants in the future. This study suggests that similar computationally host miRNA prediction strategies are warranted for identification of the miRNA targets in the other viral genomes.

Generally, the experiments are well designed and explained. The manuscript has good results and written very well.

Comments

1-     Line 39: add Family and Order of the plant [ Euphorbiaceae, Malpighiales]

2-     Line 42-43: add reference and also add information about how this virus transmit to the plant.

3-     In Figure 2: indicate the start and the end of each ORF

4-     Figure 3 should be before the title [3.2 ORF1 encoding Polyprotein.]

5-     Figure 5: it should be after line 225

6-     Figure 6: it should be after line 239

7-     Table 1: it should be after line 180

8-      In table 2: Delete lines under words and numbers.

9-     Figure 7: it should be after line 310

10- Figure 8: it should be after line 325

Author Response

Cover Letter

Dear Editor

We are grateful to the reviewers and respected editor for their insightful comments on my original research manuscript (manuscript ID# applsci-2010697) which is currently under major revision in the Applied Sciences. Coauthors and I very much appreciated the encouraging, critical and constructive comments on this manuscript by the respected editor and reviewers. The comments have been very thorough and useful in improving the manuscript. We strongly believe that the comments and suggestions have increased the scientific value of revised manuscript by many folds. We have taken them fully into account in this major revision. We are submitting the revised manuscript with the suggestion incorporated the manuscript. We have been able to incorporate changes to reflect most of the suggestions provided by the reviewers. We have highlighted the changes within the manuscript (Tracked Changes). Here is a point-by-point response to the reviewers' comments and concerns. The manuscript has been revised as per the comments given by the respected reviewers and our responses to all the comments are as follows:

RESPECTED REVIEWER 1 (GENERAL COMMENTS): 

The Manuscript [applsci-2010697] entitled (Computational Biology and Machine Learning Approaches to Identify the Rubber Tree (Hevea brasiliensis Muell.Arg.) Genome Encoded MicroRNAs Targeting Rubber Tree Virus 1) provides a first computational evidence of the reliable miRNA–mRNA interaction between specific rubber tree miRNAs and RTV1 genomic RNA transcript and offer valuable evidence for the development of RTV1-resistant rubber tree plants in the future. This study suggests that similar computationally host miRNA prediction strategies are warranted for identification of the miRNA targets in the other viral genomes.

Generally, the experiments are well designed and explained. The manuscript has good results and written very well.

General Comment1: Line 39: add Family and Order of the plant [Euphorbiaceae, Malpighiales]

Response- Respected Sir, thank you very much for your kind comment. Correction has been made in the revised manuscript. Please see line 46.

General Comment2: Line 42-43: add reference and also add information about how this virus transmits to the plant.

Response- Respected Sir, we are highly thankful to for the useful comments to improve the manuscript. It has given new direction to improve the manuscript.  Correction has been made. The new references have been added. Please see lines: 53 and references no. 3-7

General Comment3: In Figure 2: indicate the start and the end of each ORF. 

Response- Respected Sir, thank you very much for your kind comment. It has given us new direction to improve the manuscript. Correction has been made .ORF information has been added in the figure 2. Please line 163.  

General Comment4: Figure 3 should be before the title [3.2 ORF1 encoding Polyprotein.]

Response- Respected Sir, thank you very much for your kind comment. Correction has been made in the revised manuscript. Please see figure 3 and see lines 166-168

General Comment5: Figure 5: it should be after line 225.

Response- Thank you very much for your valuable comments for the improvement of our manuscript. Correction has been made. Please see figure 6 (its number change because of addition of a new figure). Please see lines 247-248.

General Comment6:  Figure 6: it should be after line 239.

Response- Thank you very much for your valuable comments for the improvement of our manuscript. Correction has been made. Please see figure 7 at lines 265-267.

General Comment7: Table 1: it should be after line 180.

Response- Thank you very much for your valuable comments for the improvement of our manuscript. Correction has been made. Table 1 has been moved. Please see lines 203-205.

General Comment8: In table 2: Delete lines under words and numbers.

Response- Thank you very much for your valuable comments for the improvement of our manuscript. Correction has been made. Please see table 2 at lines 276-277.

General Comment9: Figure 7: it should be after line 310.

Response- Thank you very much for your valuable comments for the improvement of our manuscript. Correction has been made. It has been removed as per reviewer suggestion.

General Comment10: Figure 8: it should be after line 325.

Response- Thank you very much for your valuable comments for the improvement of our manuscript. Correction has been made. Please see lines 361-362.

Reviewer 2 Report

Comments to the authors

Methods

2.9 and 2.10 can be merged. The criteria ( eg. the number/thershhold for free energy ) should be added

2.10.1 what input data and which script were used. More details need to be added in this section

Results

3.2 a table or other mthod can be used to present this results. Which would be easier to compare by the readers. The sam comment applies for section 3.3

Figure 3. From this figure, how one can see mir6167. Can it be added next to the numbers?

Figure 4. what is the numbers 0-0.7 which repeated many times ? I suggest to show this result in a way that make more sense for the miRNA position for each ORF and their common miRNAs 9if any) can be easily seen

Figure 5.  you need to add in the legend;  what level of Minmum free energy and score was considered as a potential miRNA,

Figure 6. what information can this figure deliver to the readers? I assume they are not predicted for the first time.  This data can go to sup data plus add more details in the figure legend that a reader understand which format is more likely to express/stable/ …

Table 1 is a useful data. I suggest to show the most valid/likely occurring data in colors (high/ medium and low)

Figure 7. is there any new/specific information in this figure?

Figure 8. could be an application for this study, if you could test this strategy and prove your prediction.

Figure 8 and 9 can be merged together and basically the are not provided additional data for this manuscript. Simply you could suggest they can be used for amiRNA sterategy.

Discussion

I was expecting to nominate or compare those tools/methods of miRNA target prediction. Then bringing examples that biologically/experimentally show the outcome.

Author Response

Cover Letter

Dear Editor

We are grateful to the reviewers and respected editor for their insightful comments on my original research manuscript (manuscript ID# applsci-2010697) which is currently under major revision in the Applied Sciences. Coauthors and I very much appreciated the encouraging, critical and constructive comments on this manuscript by the respected editor and reviewers. The comments have been very thorough and useful in improving the manuscript. We strongly believe that the comments and suggestions have increased the scientific value of revised manuscript by many folds. We have taken them fully into account in this major revision. We are submitting the revised manuscript with the suggestion incorporated the manuscript. We have been able to incorporate changes to reflect most of the suggestions provided by the reviewers. We have highlighted the changes within the manuscript (Tracked Changes). Here is a point-by-point response to the reviewers' comments and concerns. The manuscript has been revised as per the comments given by the respected reviewers and our responses to all the comments are as follows:

RESPECTED REVIEWER 2 COMMENTS: 

General Comment1: Methods: 2.9 and 2.10 can be merged. The criteria (e.g. the number/threshold for free energy) should be added.

Response- Thank you very much for your valuable comments for the improvement of our manuscript  We have taken reviewer’s comment in full consideration and it will be well reflected by the revised version of manuscript. Correction is made. Please see lines:  

General Comment2: 2.10.1 what input data and which script were used. More details need to be added in this section.

Response- Thank you very much for your valuable suggestion. Correction has been made. Please see lines: 138-145

General Comment3: Results: 3.2 a table or other method can be used to present these results. Which would be easier to compare by the readers? The same comment applies for section 3.3

Response- Thank you very much for your valuable suggestion. Correction has been made. We have added figure 4. Please see lines 190-192.

General Comment4: Figure 3. From this figure, how one can see hbr-miR6167. Can it be added next to the numbers?

Response- Thank you very much for your valuable suggestion. Correction has been made. Please see figure 3 at lines 166-167.

General Comment5: Figure 4. What is the numbers 0-0.7 which repeated many times? I suggest to show this result in a way that make more sense for the miRNA position for each ORF and their common miRNAs 9if any) can be easily seen

Response- Thank you very much for your valuable suggestion. Correction has been made. Please see figure 5 (Figure 4 has been changed to figure 5 because of addition of a new figure). Please see lines 222-223.

General Comment6: Figure 5.  You need to add in the legend; what level of Minimum free energy and score was considered as a potential miRNA

Response- Thank you very much for your valuable suggestion. Correction has been made as per reviewer suggestion. Legend has been added in details. Please see lines 248-253.

General Comment7: Figure 6. What information can this figure deliver to the readers? I assume they are not predicted for the first time.  This data can go to sup data plus add more details in the figure legend that a reader understand which format is more likely to express/stable/ …

Response- Thank you very much for your valuable suggestion. Correction has been made as per reviewer suggestion. Legend has been added in details. Please see lines 268-274 and supplementary table S5.

General Comment8: Table 1 is a useful data. I suggest showing the most valid/likely occurring data in colors (high/ medium and low)

Response- Thank you very much for your valuable suggestion. Correction has been made. Please table 1 at lines 203-205

General Comment9: Figure 7. is there any new/specific information in this figure?

Response- Thank you very much for your valuable suggestion. Correction has been made. Figure 7 has been removed.

General Comment10: Figure 8. Could be an application for this study, if you could test this strategy and prove your prediction.

Response- Thank you very much for your valuable suggestion. Yes, it showed a future work.

General Comment11: Figure 8 and 9 can be merged together and basically they are not provided additional data for this manuscript. Simply you could suggest they can be used for amiRNA strategy

Response- Thank you very much for your valuable suggestion. Correction has been made. Both Figures has been merged. Please see final figure 8 at lines 361-362.

General Comment12: Discussion- I was expecting to nominate or compare those tools/methods of miRNA target prediction. Then bringing examples that can biologically/experimentally show the outcome.

Response- Thank you very much for your valuable comments for the improvement of our manuscript in the discussion section.  We have taken reviewer’s comment in full consideration and it will be well reflected by the revised version of manuscript. Correction is made. Please see lines: 292, 297-300, 317-320, 324-326, 340-347.